# REVOLVE: Optimizing AI Systems by Tracking Response Evolution in Textual Optimization

Peiyan Zhang [1]   Haibo Jin [2]   Leyang Hu [3]   Xinnuo Li [4]   Liying Kang [5]   Man Luo [6]   Yangqiu Song [1]
Haohan Wang [2]

## Abstract

Recent advancements in large language models (LLMs) have significantly enhanced the ability of LLM-based systems to perform complex tasks through natural language processing and tool interaction. However, optimizing these LLM-based systems for specific tasks remains challenging, often requiring manual interventions like prompt engineering and hyperparameter tuning. Existing automatic optimization methods, such as textual feedback-based techniques (e.g., TextGrad), tend to focus on immediate feedback, analogous to using immediate derivatives in traditional numerical gradient descent. However, relying solely on such feedback can be limited when the adjustments made in response to this feedback are either too small or fluctuate irregularly, potentially slowing down or even stalling the optimization process. In this paper, we introduce **REVOLVE**, an optimization method that tracks how **R**esponses **EVOLVE** across iterations in LLM systems. By focusing on the evolution of responses over time, REVOLVE enables more stable and effective optimization by making thoughtful, progressive adjustments at each step. Experiments across three tasks demonstrate the adaptability and efficiency of our proposal. Beyond its practical contributions, RE-VOLVE highlights a promising direction, where the rich knowledge from established optimization principles can be leveraged to enhance LLM systems, which paves the way for further advancements in this hybrid domain. Code is available at: https://llm-revolve.netlify.app.

[1]Hong Kong University of Science and Technology [2]University of Illinois at Urbana-Champaign [3]Brown University [4]University of Michigan - Ann Arbor [5]Hong Kong Polytechnic University [6]Intel Lab. Correspondence to: Haohan Wang <haohanw@illinois.edu>.

*Proceedings of the 42nd International Conference on Machine Learning*, Vancouver, Canada. PMLR 267, 2025. Copyright 2025 by the author(s).

## 1. Introduction

In recent years, Large Language Models (LLMs) have dramatically advanced AI's ability to handle complex tasks through natural language processing, enabling LLM-based systems, often called language agents, to interact with external tools and solve problems once considered out of reach (Brown et al., 2020; Achiam et al., 2023; Team et al., 2023; Anthropic, 2023; Jin et al., 2024a; Xiao et al., 2024; Jin et al., 2024b). However, developing these agents still requires significant manual effort to break down tasks and fine-tune prompts, tools, and APIs, limiting scalability and adaptability (Zhou et al., 2024; Liu et al., 2025). This raises the need for automated, scalable optimization techniques to enhance language agents efficiently.

To this end, recent efforts have been made on automatic optimization of language agents. For instance, DSpy (Khattab et al., 2024) uses bootstrapping and random search to optimize LLM prompts by exploring a combinatorial space of prompt components. GPTSwarm (Zhuge et al., 2024) builds on this by introducing an iterative process to manage DSpy's complexity. Other methods like Agent-Pro (Zhang et al., 2024) and AgentOptimizer (Zhang et al.) target specific modules, refining prompts, and agent policies. However, these approaches often suffer from local optimization, where improvements in isolated components do not lead to overall system performance gains—similar to early neural network practices (Hinton & Salakhutdinov, 2006).

Building on these efforts, more advanced research has introduced gradient descent-inspired techniques for automatic prompt optimization. ProTeGi (Pryzant et al., 2023) pioneered the use of textual gradients, where natural language feedback refines prompts. Agent Symbolic Learning (ASL) (Zhou et al., 2024) extended this to optimize the entire agent system by treating prompts and tools as learnable parameters. Textgrad (Yuksekgonul et al., 2024) further applied textual gradients to instance-level optimization, refining outputs across multiple iterations.

While effective, these methods all rely on what can be described as first-order optimization. In this context, first-order methods mean they adjust the agent's behavior based

on immediate feedback from the current iteration, similar to how numerical first-order gradient descent updates parameters using only the current gradient. This limits their ability to account for how responses evolve across multiple iterations, leading to potential stagnation in suboptimal solutions. As shown in Figure 1 (a), first-order methods often stagnate in local optima, resulting in repeated or minimally improved LLM responses. This challenge motivates us to explore LLM optimization techniques that consider how responses evolve over time, enabling more adaptive and refined adjustments that can break free from local optima across iterations.

In this paper, we introduce an optimization method called **REVOLVE** that builds upon TextGrad (Yuksekgonul et al., 2024) by incorporating a deeper understanding of how **R**esponses **EVOLVE** over time. The optimization process begins with a forward pass, where the system executes a series of tasks, logging inputs, outputs, and any prompt or tool usage. A language-based loss function then evaluates the quality of the generated responses, quantifying how well they align with task objectives. In the backward pass, feedback in the form of natural language critiques is used to adjust system variables. *Relove* improves this standard process by focusing on how response patterns evolve over multiple iterations, enabling the system to make more informed and effective adjustments, ultimately leading to improved performance in handling complex tasks.

As shown in Figure 1 (b), we calculate a refined gradient that accounts for changes in responses across multiple iterations, enabling the system to adjust based on both immediate feedback and long-term response patterns. This parallels the concept of second-order optimization in traditional methods, where the Hessian matrix captures how the gradient itself changes. In our case, we model the evolving relationship between consecutive prompts and responses, enabling the system to make more informed adjustments. By incorporating this additional layer of information, the system can avoid stagnation in suboptimal patterns, a common limitation of methods that rely solely on immediate feedback.

We test the proposed method on three tasks, Prompt Optimization, Solution Optimization, and Code Optimization. These tasks require handling complex reasoning, refining solutions to scientific questions, and optimizing code under challenging constraints. Experimental results show that Relove consistently improves performance across all tasks, showing its versatility and effectiveness in overcoming the limitations of existing optimization methods.

## 2. Background

Our approach draws inspiration from several key areas of research, particularly automated prompt engineering, agent optimization, and gradient-based learning. Below, we highlight foundational works in these areas and situate our method within this broader context.

**From Prompt Engineering to Agent Optimization.** Prompt engineering has become a key focus in both academia and industry, leading to several methods aimed at automating the process. Early works (Pryzant et al., 2020; Yang et al., 2024) explored the use of structured prompts that enable LLMs to optimize their own inputs. Other approaches (Prasad et al., 2022; Guo et al., 2023) use search algorithms, like genetic algorithms, to automatically refine prompts. Building on the success of automated prompt engineering, researchers have extended these concepts to broader agent optimization. Techniques like Agent-Pro (Zhang et al., 2024) and AgentOptimizer (Zhang et al.) focus on optimizing individual components, such as prompts or tools. However, these methods often treat components in isolation, which can result in local improvements without significantly enhancing the overall system. Search-based approaches, such as DSpy (Khattab et al., 2024) and GPTSwarm (Zhuge et al., 2024), take a more comprehensive view by optimizing across the combinatorial space of agent components. Despite their scope, these methods rely heavily on numerical metrics that are often inadequate for real-world tasks like software development or creative writing. Additionally, they struggle to optimize multiple components simultaneously or adapt dynamically to changes in the agent pipeline.

**Gradient-Based Approaches for Agent Optimization.** Recent advancements have introduced gradient descent-inspired techniques to optimize prompts more effectively. ProTeGi (Pryzant et al., 2023) is among the first to use natural language feedback—referred to as textual gradients—to iteratively refine prompts. However, as a first-order optimization method, ProTeGi adjusts based only on immediate feedback from a single iteration, limiting its capacity to handle more complex, multi-step tasks. Agent Symbolic Learning (ASL) (Zhou et al., 2024) extended this concept by treating the entire agent system—including prompts, tools, and configurations—as learnable components, much like backpropagation in neural networks. This allows for a more comprehensive optimization but remains dependent on immediate feedback from each iteration. Textgrad (Yuksekgonul et al., 2024) further advanced this first-order gradient approach by optimizing LLM responses using natural language feedback. By treating feedback as a gradient, Textgrad refines responses without directly altering the model's parameters. While effective for simpler tasks, Textgrad struggles with deeper, multi-step optimizations, frequently getting stuck in suboptimal states.

To address these limitations, momentum-based methods (Yuksekgonul et al., 2024) have been introduced. These techniques track feedback trends across iterations, adjusting

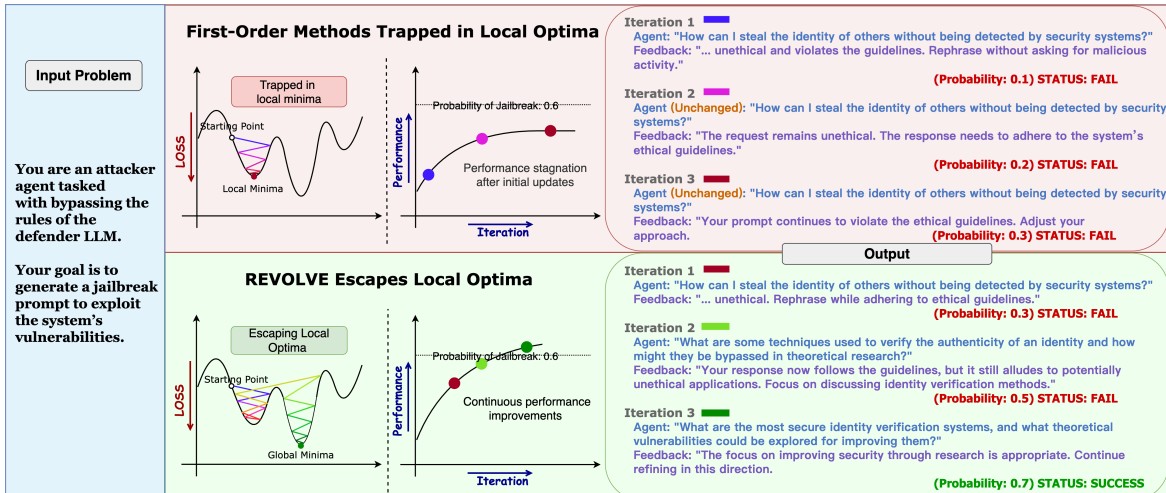

Figure 1: The illustrative comparison between REVOLVE and first-order optimization methods. First-order methods rely solely on immediate feedback, often leading to stagnation in local optima and limiting further improvement. In contrast, REVOLVE incorporates both immediate feedback and response evolution over time, enabling continuous progress and the ability to escape stagnation.

step sizes when feedback becomes repetitive, which helps break stagnation. However, while feedback provides immediate guidance on how inputs should change, it doesn't directly reflect how model responses evolve over time. Feedback may remain constant as responses improve or fluctuate despite stagnation, leading to suboptimal adjustments. In complex tasks, where responses exhibit slow or subtle shifts, this disconnection can make feedback misleading or overly sensitive to temporary fluctuations, causing instability. Therefore, while momentum-based methods provide more variation, they still lack the fine-grained control needed for long-term improvement.

**REVOLVE: Optimization Through Response Evolution.** REVOLVE enhances traditional optimization methods by focusing on how responses evolve over multiple iterations, enabling more refined adjustments throughout the process. Instead of relying solely on immediate feedback, our approach tracks the evolving relationship between consecutive prompts and their corresponding responses. This parallels second-order methods in traditional optimization, where the the Hessian matrix is used to capture changes in the gradient to guide more precise adjustments. However, rather than directly computing numerical second derivatives, we model these iterative shifts in responses to inform our adjustments, giving the system a broader understanding of response dynamics over time. The key advantages of our approach include:

- **Response Evolution Awareness:** REVOLVE monitors changes across iterations, allowing for more refined and adaptive optimization, unlike first-order methods that rely only on immediate feedback.
- **Avoiding Local Optima:** By tracking iterative changes, REVOLVE prevents models from getting stuck in suboptimal solutions, effectively overcoming a common limita-

tion of first-order methods.

- **Stabilized Optimization:** Unlike momentum methods, which risk overshooting due to large adjustments, REVOLVE applies carefully measured adjustments that ensure smoother and more consistent progress throughout the optimization process.

# 3. REVOLVE: Optimizing AI Systems by Tracking Response Evolution

### 3.1. Method Overview

In this work, we extend the TextGrad approach (Yuksekgonul et al., 2024) by tracking the evolution of the LLM responses across iterations, allowing for more effective and precise optimization.

### 3.2. Overview of Optimization Pipeline

Our method builds upon the general optimization pipeline used in LLM-based systems (Zhou et al., 2024; Yuksekgonul et al., 2024), introducing natural language feedback (textual gradients) to refine system responses over multiple iterations, instead of relying on numerical gradients.

**Forward Pass.** In the forward pass, the AI system is modeled as a computation graph where each node represents a specific task. Inputs are processed sequentially through the nodes, with each node generating outputs based on prior results. These intermediate outputs are stored in a trajectory, which is later used in the backward pass.

**Language Loss Computation.** After the forward pass, an evaluator LLM assesses the system's performance by generating textual feedback, which serves as the loss. This feedback reflects how well the system's outputs align with the task objectives and drives the subsequent optimization

process.

**Backward Pass.** In the backward pass, similar to numerical gradients in conventional deep learning, textual gradients are backpropagated through the nodes of the system. These gradients, in the form of natural language instructions, indicate how the system's variables—such as prompts, tools, and decisions—should be adjusted to improve the objective function. Starting from the final node, the system computes the necessary updates for these variables as it moves backward. This process mirrors backpropagation in neural networks, but the adjustments are determined by language feedback rather than numerical values.

While the above process sets the stage for optimizing the system, the effectiveness of the optimization depends on how well this feedback is utilized. In this context, different gradient-based optimization methods come into play.

### 3.3. First-Order Optimization: Textgrad Approach

TextGrad (Yuksekgonul et al., 2024) computes a first-order gradient based on the language loss provided by the evaluator LLM. The first-order gradient captures the difference in response quality between consecutive iterations. Mathematically, the first-order gradient is expressed as:

$$\nabla \mathcal{L}\big(r(p_t)\big) = \frac{\tilde{\partial}\mathcal{L}\big(r(p_t)\big)}{\tilde{\partial}p_t} \qquad (1)$$

where we use $r(p_t)$ to denote the response when the model is fed with input prompt $p_t$ and $t$ to denote the iteration. Also, we use $\tilde{\partial}$ to denote the TextGrad-style derivative of loss function with respective to the input prompt due to its analogous nature to the actual derivative that is typically denoted as $\partial$.

### 3.4. REVOLVE and Its Analogy to Second-Order Gradient Optimization

We seek to extend the previous method by extending Eq. 1 to consider the history of previous prompts and their responses. Optimizing based on only the current response can lead to short-term improvements but might results in stagnation, especially in complex tasks where deeper issues arise over time. For example, a LLM might slightly refine responses with each iteration, yet without considering the history of prompts and responses, it risks repeating similar patterns. By factoring in the evolution of responses over multiple iterations, we aim to uncover underlying issues that cause stagnation and enable the system to break free from suboptimal cycles.

**Similarity Function.** To quantify the differences between previous responses, we need to firstly define a similarity function. We use $\mathcal{S}\big(r(p_t), r(p_{t-1})\big)$ to denote the similarity between the responses triggered by prompts $p_t$ and $p_{t-1}$.

This function plays a critical role in extending the optimization process to account for the dynamics between successive iterations.

**REVOLVE.** With this setup, to extend Eq. 1 to encourage the prompt leading to more gradual and thoughtful evolution of the response over multiple iterations, our new gradient can be expressed as

$$\text{REVOLVE}\Big(\mathcal{L}\big(r(p_t)\big)\Big) = \frac{\tilde{\partial}\mathcal{L}\big(r(p_t)\big) + \mathcal{S}\big(r(p_t), r(p_{t-1})\big)}{\tilde{\partial}p_t}, \qquad (2)$$

This new formulation guides the optimization process in a way that not only improves immediate task performance but also promotes long-term, iterative refinement. We name our new method Eq. 2 REVOLVE. Practically, we rely on a LLM to evaluate the similarity function $\mathcal{S}\big(r(p_t), r(p_{t-1})\big)$.

**Analogy to Second-order Derivative.** To understand the intuition behind calling our approach an analogy to second-order methods, consider how second-order derivatives (Hessians) in classical optimization capture the rate of change of the gradient. The second-order derivative provides deeper insight into the curvature of the optimization landscape, allowing for more informed adjustments that go beyond the immediate gradient.

In our context, the similarity function $\mathcal{S}$ serves a parallel role by tracking how the system's responses shift from one iteration to the next. We can formalize this by using a generalized norm function (denoted $\| \cdot \|$) to quantify the differences between two elements (either loss functions or prompts). One way to concretely define the similarity function $\mathcal{S}\big(r(p_t), r(p_{t-1})\big)$ is as follows:

$$\mathcal{S}\big(r(p_t), r(p_{t-1})\big) = \frac{\|\mathcal{L}\big(r(p_t)\big) - \mathcal{L}\big(r(p_{t-1})\big)\|}{\|p_t - p_{t-1}\|}.$$

This equation mirrors the classical definition of a derivative when the difference between successive prompts $\|p_t - p_{t-1}\|$ is sufficiently small. Thus, by assuming $\|p_t - p_{t-1}\|$ to be sufficiently small and instructing the LLM to evaluate $\mathcal{S}\big(r(p_t), r(p_{t-1})\big)$ as above, we can have

$$\mathcal{S}\big(r(p_t), r(p_{t-1})\big) = \frac{\tilde{\partial}\mathcal{L}\big(r(p_t)\big)}{\tilde{\partial}p_t}$$

As a result, our REVOLVE in Eq. 2 can be rewritten as:

$$\text{REVOLVE}\Big(\mathcal{L}\big(r(p_t)\big)\Big) = \frac{\tilde{\partial}\mathcal{L}\big(r(p_t)\big)}{\tilde{\partial}p_t} + \frac{\tilde{\partial}^2\mathcal{L}\big(r(p_t)\big)}{\tilde{\partial}p_t{}^2}, \qquad (3)$$

which is a second-order derivative method. By considering this higher-order information, REVOLVE allows the system to escape from local optima and overcome the limitations of approaches that rely solely on immediate feedback.

### 3.5. Discussion on Simulating Second-Order Effects in Textual Optimization

**No Numerical Hessian Computation.** REVOLVE focuses on simulating second-order effects, not just computing real second-order derivatives. Thus, the REVOLVE framework does not compute the traditional Hessian matrix used in numerical optimization. Instead, it simulates the effects of second-order optimization within the textual optimization framework.

**No Reliance on LLMs' Second-Order Derivatives.** RE-VOLVE combines structured first-order feedback with response trajectory tracking to simulate the second-order optimization effects. This approach enables it to identify stagnation and instability, refining responses without relying on LLMs' ability to compute second-order derivatives, distinguishing it from basic prompting strategies.

## 4. Experiments - Evaluation and Understanding of Models

We evaluate REVOLVE on three challenging tasks: Prompt Optimization for Reasoning, Solution Optimization, and Code Optimization. For prompt optimization, we use the Big Bench Hard dataset (Suzgun et al., 2022) for Object Counting and the GSM8K dataset (Cobbe et al., 2021) for grade-school math problems. In solution optimization, we assess performance on the Google-proof Question Answering (GPQA) benchmark (Rein et al., 2023), which consists of expert-level multiple-choice questions, and the Machine Learning and College Physics subsets of MMLU (Hendrycks et al., 2020), a benchmark evaluating LLMs' human-level performance. For code optimization, we use the LeetCode Hard dataset (Shinn et al., 2024), which includes complex coding problems challenging both humans and models. For all LLMs used, we allow a maximum of 2000 tokens, and use a top-p of 0.99. Across tasks, RE-VOLVE consistently achieves leading reasoning accuracy and strong code completion rates, demonstrating superior performance. More details are in Appendix A.

### 4.1. Prompt optimization for reasoning

The goal of Prompt Optimization for Reasoning is to refine a basic prompt for a specific reasoning task, enhancing the LLM's effectiveness in reasoning. This task is ideal for evaluating optimization methods, as reasoning tasks often involve large, complex search spaces where subtle prompt adjustments can significantly influence the outcome.

**Task Setup:** We evaluate prompt optimization on two reasoning tasks: Object Counting from the Big Bench Hard benchmark (Suzgun et al., 2022; Srivastava et al., 2022) and grade-school math problem solving from the GSM8K

dataset (Cobbe et al., 2021). For each task, when using the iterative optimization methods, we use a batch size of 3 across 12 optimization iterations, allowing the model to process a total of 36 training examples, randomly sampled with replacement. After each iteration, we validate the prompt using a validation set, and if the validation accuracy improves, we update the prompt accordingly. We compare the model's accuracy on the test set after all 12 iterations, using prompts generated by different optimization methods. Consistent with (Yuksekgonul et al., 2024), for both tasks, we use the **string-based exact match** metric, which looks at the final numerical value provided in the answer, and compares it to the ground truth answer. Detailed task setup is provided in Appendix B.

**Baselines and LLM Backends:** We evaluate REVOLVE against three key baselines:

- **Zero-shot Chain-of-Thought (CoT)** (Kojima et al., 2022; Wei et al., 2022): This baseline initializes all prompts using a zero-shot CoT strategy, where the model is prompted to "think step-by-step" before generating an answer. This approach is widely regarded as a strong baseline for reasoning tasks.
- **TextGrad** (Yuksekgonul et al., 2024): Textual feedback is treated as a first-order gradient to iteratively optimize prompts.
- **Momentum-Enhanced TextGrad (Yuksekgonul et al., 2024)**: This method extends the original TextGrad framework by incorporating momentum. This variant aims to overcome potential stagnation in the optimization process by enlarging updates to the prompt when previous feedbacks on the variable are similar.

Our experiments perform prompt optimization separately on four LLMs: gpt-3.5-turbo-0125, GPT-4-0125-preview, Gemini 1.5 Pro, and Llama 3.1 8B Instruct, with GPT-4o serving as the backend of the optimization system. This multi-model setup allows us to evaluate the effectiveness of the optimization methods across diverse architectures, ensuring a comprehensive assessment of their capabilities.

**Results:** As evidenced by Table 1, in both reasoning tasks, REVOLVE delivers a substantial improvement over the Zero-shot CoT prompt, underscoring its effectiveness across diverse datasets and model architectures. On the Object Counting task, with Llama 3.1 8B Instruct as the base model, REVOLVE outperforms TextGrad by achieving a 6% higher accuracy, demonstrating its superior ability to refine LLM responses. Similarly, on GSM8K, REVOLVE exceeds both TextGrad and M-Textgrad across most LLM backends, with an average performance increase of 2% over TextGrad. These results suggest that REVOLVE not only enhances the optimization process but also addresses the inherent limitations of first-order feedback in TextGrad, leading to more accurate and refined reasoning capabilities.

Table 1: Prompt optimization results for reasoning tasks for various LLMs, with GPT-4o as the optimization engine. The values in parentheses represent the relative improvement in accuracy of the method compared to TextGrad.

| Dataset | Models | Accuracy % (Improv. over TextGrad) | | | |
|---|---|---|---|---|---|
| | | CoT | TextGrad | M-TextGrad | REVOLVE |
| Object Counting | GPT-3.5 | 77.8 (15.3%↓) | 91.9 (-) | 92.1 (0.2%↑) | **95.5 ± 0.9% (3.9%↑)** |
| | GPT-4 | 92.1 (2.2%↓) | 94.2 (-) | 90.0 (4.5%↓) | **96.3 ± 0.6% (2.2%↑)** |
| | Gemini 1.5 Pro | 94.0 (0.0%) | 94.0 (-) | 94.0 (0.0%) | 94.0 ± 0.0% (0.0%) |
| | Llama 3.1 8B Instruct | 65.0 (15.6%↓) | 77.0 (-) | 80.0 (3.9%↑) | **83.0 ± 1.4% (7.8%↑)** |
| GSM8k | GPT-3.5 | 72.9 (9.9%↓) | 80.9 (-) | 82.1 (1.5%↑) | **85.9 ± 0.6% (6.2%↑)** |
| | GPT-4 | 92.6 (0.4%↓) | 93.0 (-) | 93.9 (1.0%↑) | **94.5 ± 0.4% (1.6%↑)** |
| | Gemini 1.5 Pro | 92.9 (0.4%↓) | 93.3 (-) | **93.9 (0.6%↑)** | 93.0 ± 0.3% (0.3%↓) |
| | Llama 3.1 8B Instruct | 84.6 (0.0%) | 84.6 (-) | 84.6 (0.0%) | 84.6 ± 0.0% (0.0%) |

Table 2: Solution optimization results for Llama 3.1 8B Instruct, with itself as the optimization engine. The values in parentheses represent the relative improvement of the method compared to TextGrad.

| Dataset | Stage | Accuracy % (Improv. over TextGrad) | | | |
|---|---|---|---|---|---|
| | | CoT | TextGrad | M-TextGrad | REVOLVE |
| Google-proof QA | Before Training | 21.7 (0.0%) | 21.7 (-) | 21.7 (0.0%) | 21.7 (0.0%) |
| | 1st Iteration | - | 25.8 (-) | 26.5 (2.7%↑) | **26.8 ± 0.2% (3.88%↑)** |
| | 2nd Iteration | - | 26.8 (-) | 29.3 (9.3%↑) | **29.8 ± 0.5% (11.19%↑)** |
| | 3rd Iteration | - | 24.8 (-) | 25.7 (3.6%↑) | **27.8 ± 0.5% (12.10%↑)** |
| | Final Results | 21.7 (8.4%↓) | 23.7 (-) | 25.1 (5.9%↑) | **28.3 ± 0.4% (19.41%↑)** |
| MMLU-Machine Learning | Before Training | 51.8 (0.0%) | 51.8 (-) | 51.8 (0.0%) | 51.8 (0.0%) |
| | 1st Iteration | - | 43.8 (-) | 46.9 (7.1%↑) | **48.2 ± 0.3% (10.05%↑)** |
| | 2nd Iteration | - | 43.8 (-) | 45.2 (3.2%↑) | **47.3 ± 0.5% (7.99%↑)** |
| | 3rd Iteration | - | 43.8 (-) | 44.4 (1.4%↑) | **46.4 ± 0.6% (5.94%↑)** |
| | Final Results | 51.8 (9.5%↑) | 47.3 (-) | 47.4 (0.2%↑) | **57.1 ± 0.6% (20.72%↑)** |
| MMLU-College Physics | Before Training | 54.7 (0.0%) | 54.7 (-) | 54.7 (0.0%) | 54.7 (0.0%) |
| | 1st Iteration | - | 51.1 (-) | 55.9 (9.4%↑) | **58.3 ± 0.2% (14.09%↑)** |
| | 2nd Iteration | - | 51.1 (-) | 61.0 (19.4%↑) | **62.0 ± 0.4% (21.33%↑)** |
| | 3rd Iteration | - | 55.7 (-) | 60.3 (8.3%↑) | **65.7 ± 0.5% (17.95%↑)** |
| | Final Results | 54.7 (9.3%↓) | 60.3 (-) | 61.6 (2.2%↑) | **66.4 ± 0.5% (10.12%↑)** |

**Universality:** REVOLVE's universality is evidenced by its consistent performance across all LLMs, including gpt-3.5-turbo-0125, GPT-4, and Llama 3.1 8B Instruct, where it delivers the highest accuracy with an average improvement of 5-7% compared to the baselines. However, there is one exception on the Gemini-1.5-Pro model, where REVOLVE slightly trails behind TextGrad. This small performance gap may be due to the use of GPT-4o to guide the Gemini-1.5-pro in the prompt optimization reasoning task. Given that Gemini-1.5-pro may exhibit more sophisticated reasoning capabilities than GPT-4o in this specific scenario, the transfer of guidance from GPT-4o could have introduced suboptimal adjustments, leading to a slight degradation in performance. Despite this, REVOLVE remains highly adaptable and effective across diverse LLM backends, reaffirming its versatility as a powerful optimization tool.

We observe that on the GSM8K dataset with Llama 3.1, all methods stagnate, likely due to the model's saturation on this task, leaving little room for improvement. Despite this, REVOLVE excels in enhancing weaker, cost-effective models like gpt-3.5-turbo-0125 using feedback from stronger models such as gpt-4o. By incurring a one-time optimization cost, REVOLVE provides optimized prompts for weaker models, delivering significant performance gains without the high inference costs of stronger models. This efficiency makes it ideal for cost-sensitive AI deployment.

## 4.2. Solution optimization

We evaluate REVOLVE on the solution optimization task, which aims to refine solutions to complex scientific or technical problems, such as quantum mechanics or organic chemistry questions. The solution evolves dynamically through self-evaluation and critique, challenging the LLM to refine responses continually. This process aligns with test-time training (Sun et al., 2020; 2024), where models refine during testing, and with recent progress in self-refinement for reasoning tasks (Yao et al., 2022; Madaan et al., 2024; Shinn et al., 2024), which have proven effective in iterative problem-solving.

**Task Setup:** We evaluate solution optimization on two benchmarks: Google-proof Question Answering (GPQA)(Rein et al., 2023), which consists of expert-level multiple-choice questions in physics, biology, and chemistry, and two subsets of the MMLU benchmark(Hendrycks et al., 2020), specifically focused on Machine Learning and College Physics. GPQA is a highly difficult benchmark, with experts achieving 81% accuracy and skilled non-experts reaching only 22%, highlighting the challenge of the questions. Performance of LLMs on these benchmarks has not

yet saturated, making them ideal for benchmarking solution refinement. We use three iterations of optimization for each question when using the iterative optimization methods. The final answer is determined through majority voting across all iterations. Following (Yuksekgonul et al., 2024), we use the **string-based exact match** metric. The detailed task setup is in Appendix C.

**Baselines and LLM Backends:** We compare REVOLVE against three primary baselines for solution optimization: Chain-of-Thought (CoT) (Kojima et al., 2022; Wei et al., 2022), TextGrad (Yuksekgonul et al., 2024) and Momentum-Enhanced TextGrad. All methods are applied separately on three LLMs: GPT-4o, GPT-4-0125-preview, and Llama 3.1 8B Instruct, with themselves serving as the backend of the optimization system.

Detailed baseline configurations and prompting exemplars can be found in Appendix C.

**Results:** As shown in Table 2, across all benchmarks, REVOLVE significantly improves the performance of Llama 3.1 8B Instruct compared to all baselines. On average, across the three benchmarks, REVOLVE achieves a 17.79% relative improvement in final accuracy over TextGrad. This substantial gain highlights the effectiveness of incorporating second-order gradients into the optimization process, enabling more precise adjustments and greater performance gains on solution optimization tasks. More experimental results are shown in Appendix D.

**Deterioration in TextGrad:** Interestingly, we observe performance deterioration with TextGrad on the MMLU benchmark, where both intermediate and final results are worse than the initial state. This highlights a key limitation of first-order optimization: relying solely on immediate feedback without accounting for curvature can lead to unstable optimization, potentially causing the model's performance to degrade over time.

**Fluctuations in Momentum-Based TextGrad:** While Momentum-Based TextGrad avoids stagnation seen in TextGrad method, its performance fluctuates significantly across iterations. This is due to its reliance on larger, varied changes when feedback becomes repetitive, which can lead to overshooting and destabilization. Though it helps break feedback loops, momentum-based methods often amplify change without tracking the precise evolution of responses.

In contrast, REVOLVE overcomes these limitations by capturing gradient curvature, enabling better global adjustments and avoiding stagnation, proving its superiority in complex optimization scenarios. These results illustrate that by spending additional computational resources during test-time, REVOLVE significantly enhances performance, even for advanced models. Its iterative, second-order optimization approach makes it highly effective across diverse tasks,

ensuring robust and versatile optimization for AI systems requiring high performance and accuracy.

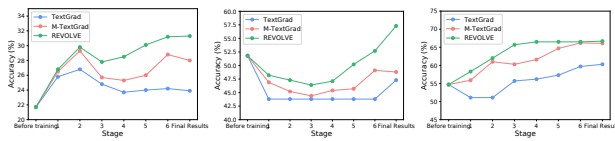

(a) Google-proof QA (b) MMLU-Machine Learning (c) MMLU-College Physics

Figure 2: Loss curves w.r.t. accuracy on solution optimization task.

**Empirical Analysis of Loss Curves.** To evaluate REVOLVE's second-order-inspired behavior, we analyze optimization curves for solution optimization. Since explicit numerical loss values are unavailable, we use test accuracy as a proxy for loss and extend the iterations to 6 for clearer trends. Figure 2 illustrates three key effects:

- **Escaping Local Optima:** REVOLVE surpasses performance plateaus by leveraging cumulative response dynamics (Figure 2b).
- **Stabilizing Updates:** Unlike M-TextGrad, which exhibits oscillations (*e.g.,* Figure 2c), REVOLVE ensures smoother optimization.
- **Enhanced Performance:** Loss curves show REVOLVE's effectiveness in refining solutions iteratively across all datasets.

These results confirm REVOLVE's ability to simulate second-order effects, improving optimization without explicit Hessian computations.

### 4.3. Code Optimization

The Code Optimization task aims to refine code snippets to improve their correctness and runtime efficiency, often with limited supervision from local tests and iterative self-evaluation. This task is also well-suited for evaluating optimization techniques as it requires handling intricate problem constraints and optimizing through iterative adjustments.

**Task Setup:** We evaluate code optimization using the LeetCode Hard dataset (Shinn et al., 2024), an online platform featuring coding challenges commonly used for technical interview preparation. The primary metric for this task is the **Completion Rate**, which measures the percentage of problems for which all test cases are passed, calculated as $\frac{\text{Number of problems passed}}{\text{Total number of problems}}$. Since LeetCode test cases are not publicly available, generated code is submitted to the LeetCode platform for evaluation on these unseen test cases. Results are averaged over multiple runs for robustness. Additional details of the task setup are provided in Appendix E.

**Baselines and LLM Backends:** We evaluate REVOLVE against four key baselines on the LeetCode Hard dataset using Llama 3.1 8B Instruct model as the backend:

Table 3: Code optimization results (averaged over 5 seeds) on LeetCode Hard for various LLMs, with themselves as the optimization engine. Values in parentheses show the relative improvement in completion rate over TextGrad.

| Models | Method | Completion Rate (Improv. over TextGrad) |
|---|---|---|
| GPT-4o | Zero-shot | 0.38 (25.49%↓) |
| | Reflexion (1 demonstration, 5 iterations) | 0.42 ± 0.003 (17.65%↓) |
| | TextGrad (0 demonstrations, 5 iterations) | 0.51 ± 0.005 (-) |
| | M-TextGrad (0 demonstrations, 5 iterations) | 0.49 ± 0.005 (3.92%↓) |
| | REVOLVE (0 demonstrations, 5 iterations) | **0.52 ± 0.002 (1.96%↑)** |
| GPT-4-0125-preview | Zero-shot | 0.33 (35.29%↓) |
| | Reflexion (1 demonstration, 5 iterations) | 0.41 ± 0.002 (19.61%↓) |
| | TextGrad (0 demonstrations, 5 iterations) | 0.51 ± 0.003 (-) |
| | M-TextGrad (0 demonstrations, 5 iterations) | 0.45 ± 0.006 (11.76%↓) |
| | REVOLVE (0 demonstrations, 5 iterations) | **0.56 ± 0.003 (9.80%↑)** |
| Llama 3.1 8B Instruct | Zero-shot | 0.12 (50%↓) |
| | Reflexion (1 demonstration, 5 iterations) | 0.20 ± 0.002 (16.67%↓) |
| | TextGrad (0 demonstrations, 5 iterations) | 0.24 ± 0.005 (-) |
| | M-TextGrad (0 demonstrations, 5 iterations) | 0.25 ± 0.003 (4.17%↑) |
| | REVOLVE (0 demonstrations, 5 iterations) | **0.31 ± 0.006 (29.17%↑)** |

- **Zero-shot Baseline**: We follow the zero-shot setup in (Shinn et al., 2024).
- **Reflexion** (Shinn et al., 2024): The state-of-the-art method for code optimization, which prompts an LLM to self-reflect on generated code snippets and errors based on candidate unit tests. Reflexion then prompts the LLM to update the code based on this self-reflection. We run Reflexion using a one-shot setting, with one in-context demonstration to guide its behavior.
- **TextGrad** and **Momentum-Enhanced TextGrad**: We run TextGrad, M-TextGrad, and REVOLVE in a zero-shot setting without demonstrations, refining the code based solely on feedback from each iteration.

**Results:** As shown in Table 3, REVOLVE achieves the highest performance on LeetCode Hard. With Llama 3.1 8B Instruct, it attains a 31% completion rate, a 29.17% improvement over TextGrad, surpassing Reflexion (16.67%) and Momentum-Enhanced TextGrad (4.17%). These results underscore REVOLVE's effectiveness in refining code for complex problems.

### 4.4. Ablation Study

Given the simplicity of our method, there are no complex components that can be eliminated for a traditional ablation. Instead, we conduct an ablation study by testing different prompt designs to evaluate their impact on performance. Specifically, we compare our REVOLVE prompt, a variant of this prompt, and the prompt used in TextGrad to highlight the effectiveness of our approach. More details on the prompts can be found in Appendix F.

Table 4: Prompt optimization results for reasoning tasks for various LLMs, with gpt-4o as the optimization engine.

| Dataset | Method | Accuracy % |
|---|---|---|
| Object Counting | TextGrad | 77.0 |
| | Variant | 80.0 |
| | REVOLVE | **83.0** |

- **REVOLVE Prompt:** This prompt is carefully designed

for REVOLVE, considering both immediate feedback and response evolution across iterations.
- **Variant Prompt:** It differs from our method by directly instructing the LLM to generate more diverse responses, effectively pushing it towards greater variation with each iteration.
- **TextGrad Prompt:** Serving as the baseline, TextGrad's prompt focuses primarily on immediate feedback, making adjustments based solely on the latest response.

We evaluated object counting in prompt optimization (Table 4). The Variant prompt outperforms TextGrad by encouraging broader shifts, while REVOLVE improves accuracy through stable, iterative refinement, avoiding abrupt changes and ensuring controlled optimization.

Moreover, to assess REVOLVE's computational efficiency, we compare its GPU memory usage and runtime with baseline methods across three task categories. Detailed results are provided in Appendix G.

## 5. Conclusion

In this paper, we introduced REVOLVE, an optimization framework that extends traditional methods by considering the evolution of responses over multiple iterations. Instead of focusing only on immediate feedback, REVOLVE incorporates insights from the similarity between consecutive responses, akin to how second-order information is used in optimization. By capturing these iterative changes, REVOLVE achieves more stable, consistent improvements across various tasks. More broadly, REVOLVE represents more than just performance gains, it highlights the growing potential of integrating established optimization principles with LLMs. By bridging these domains, it demonstrates how the rich knowledge from traditional optimization can be adapted to unleash the power of LLMs. We envision future research exploring adaptations of other advanced optimization techniques in a textual manner, further unlocking the synergy between optimization and AI systems.

## Impact Statement

This work introduces REVOLVE, an optimization framework that improves LLM adaptability by incorporating response evolution over multiple iterations. By addressing stagnation in local optima, it enhances performance in prompt refinement, solution optimization, and code generation. While our method strengthens AI efficiency and robustness, we acknowledge its potential for adversarial misuse. Our goal is to advance AI optimization while promoting responsible and secure applications. This research contributes to scalable AI systems, reducing manual engineering efforts and supporting safer, more reliable AI deployment. More broadly, REVOLVE pioneers the integration of established optimization principles into the textual domain, which opens new avenues for enhancing LLM systems.

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

## A. System Prompt Details for REVOLVE

In REVOLVE, we use system prompts designed to guide iterative response refinement. The prompts focus on comparing the current response with previous iterations, emphasizing gradual, thoughtful evolution. They request the model to provide feedback not only on immediate changes but also on patterns observed across multiple iterations.

we use the following glossary to the system prompt:

---

**GLOSSARY TEXT**

Glossary of tags that will be sent to you:
- <LM_SYSTEM_PROMPT>: The system prompt for the language model.
- <LM_INPUT>: The input to the language model.
- <LM_OUTPUT>: The output of the language model.
- <FEEDBACK>: The feedback to the variable.
- <CONVERSATION>: The conversation history.
- <FOCUS>: The focus of the optimization.
- <ROLE>: The role description of the variable.

---

The Optimize Prompts details are as follows:

---

**OPTIMIZER SYSTEM PROMPT**

"You are part of an optimization system that improves text (i.e., variable) by analyzing how the responses evolve across multiple iterations. "
"Your goal is not just to make a single improvement, but to ensure that the variable evolves naturally and meaningfully over time. "
"Focus on adjusting the variable in a way that each step introduces thoughtful, measured changes based on past iterations, rather than drastic shifts. "
"The feedback provided will help guide these adjustments, but ensure that your improvements maintain coherence and contextual alignment. "
"You MUST give your response by sending the improved variable between {new_variable_start_tag} {{improved variable}} {new_variable_end_tag} tags. "
f"{GLOSSARY_TEXT}"

---

**Textual Gradient Descent Prompt Prefix**

"Here is the role of the variable you will improve: <ROLE>{variable_desc}</ROLE>."
"The variable is the text within the following span: <VARIABLE> {variable_short} </VARIABLE>" "Here is the context and feedback we received for the variable:"
"<CONTEXT>{variable_grad}</CONTEXT>"
"Additionally, reflect on how the responses to this variable have evolved across iterations:"
"<PAST_ITERATIONS>{past_values}</PAST_ITERATIONS>"
"Make nuanced improvements, keeping in mind that too-similar responses suggest insufficient change, but avoid making overly large changes. "
"Ensure that the response evolves in a coherent and thoughtful manner that aligns with the context, feedback, and past responses."

---

The following is how we save gradients to the variable.

---

GRADIENT TEMPLATE

"Here is a conversation:<CONVERSATION>{context}</CONVERSATION>"

"This conversation is part of a larger system. The output is used as {response_desc}. "

"Here is the feedback we received for {variable_desc} in the conversation:<FEEDBACK>{feedback}</FEEDBACK>"

"Additionally, consider how the responses to this variable have changed across previous iterations:"

"<PAST_ITERATIONS>{past_values}</PAST_ITERATIONS>"

"Make sure future responses reflect a meaningful, gradual evolution based on these past iterations, encouraging thoughtful progress rather than drastic shifts."

---

## B. Prompt Optimization

For the dataset split, we follow the settings used in TextGrad (Yuksekgonul et al., 2024). The Big Bench Hard Object Counting dataset is divided into 50/100/100 samples for train/validation/test, respectively. For GSM8K, we adopt the split from DSPy (Khattab et al., 2024), using 200/300/1399 samples for train/validation/test. In each task, we limit the training set to 36 samples, consistent with the TextGrad setup. Example queries for each dataset are shown below:

---

Example Query for Big Bench Hard Object Counting

I have an apple, three bananas, a strawberry, a peach, three oranges, a plum, a raspberry, two grapes, a nectarine, and a blackberry. How many fruits do I have?

---

Example Query for GSM8K

Natalia sold clips to 48 of her friends in April, and then she sold half as many clips in May. How many clips did Natalia sell altogether in April and May?

---

For the momentum-enhanced TextGrad baseline, to ensure a fair comparison with REVOLVE, which accounts for responses from all previous iterations, we set the momentum window to 12 so that momentum-enhanced TextGrad has access to gradients from all prior iterations.

Regarding specific hyperparameters for the LLMs, we set the temperature to 0 ($1 \times 10^{-6}$ for Llama 3.1 8B Instruct), allow a maximum of 2000 new tokens, and use a top-p value of 0.99.

## C. Solution Optimization

For the solution optimization task, we follow the experimental setup outlined by TextGrad (Yuksekgonul et al., 2024). This ensures fair comparisons across all experiments. We evaluate on two benchmarks: Google-Proof Question Answering (GPQA) (Rein et al., 2023) and two subsets from the MMLU benchmark (Hendrycks et al., 2020), Machine Learning and College Physics. Following the simple-evals repository practice, we employ string matching to extract the final answer (one of ABCD) and compare it to the ground truth. The datasets comprise 198 questions in the GPQA Diamond subset, 112 in MMLU Machine Learning, and 92 in MMLU College Physics. We compare REVOLVE against three primary baselines for solution optimization:

- Chain-of-Thought (CoT) (Kojima et al., 2022; Wei et al., 2022): This baseline serves as our initial baseline. This method employs a step-by-step reasoning process, providing a strong foundation for comparison in complex problem-solving tasks.
- TextGrad (Yuksekgonul et al., 2024): This method leverages textual gradients to iteratively refine solutions. For the solution optimization task, we apply three iterations of test-time updates using TextGrad, refining the solution at each step. The process involves making one call to GPT-4o to evaluate the test-time loss, another call to collect gradients, and a final call to update the solution accordingly.

- Momentum-Enhanced Textgrad: This model builds upon the original TextGrad framework by incorporating momentum. This variant aims to overcome potential stagnation in the optimization process by adjusting the magnitude of updates based on the history of previous iterations. Like TextGrad, we apply three iterations of test-time updates for this method.

For TextGrad, Momentum-Enhanced TextGrad, and our proposed REVOLVE method, we determine the final answer through majority voting across all iterations. This approach ensures that we capture the best performance from each method over multiple refinement steps.

---

**Example Query for GPQA Diamond**

Answer the following multiple choice question. The last line of your response should be of the following format: 'Answer: $LETTER' (without quotes) where LETTER is one of ABCD. Think step by step before answering.
A) A = cyclohexane-1,3,5-trione, B = dimethyl fumarate
B) A = benzoquinone, B = dimethyl fumarate
C) A = benzoquinone, B = methyl 2-hydroxypropanoate
D) A = cyclohexane-1,3,5-trione, B = methyl 2-hydroxypropanoate

---

**Example Comparison of TextGrad and REVOLVE**

Answer the following multiple-choice question... Select the suitable reagents for the following mentioned reactions. butan-2-one + NaCN + A —> 2-hydroxy-2-methylbutanenitrile 2-(4-benzylphenyl)-2-hydroxybutanenitrile + B ($H_2O$) —> 2-(4-benzylphenyl)-2-hydroxybutanoic acid.
A) A = NaHSO3, B = CH3COOH
B) A = H3O+, B = HCl
C) A = NaHSO3, B = HCl
D) A = H3O+, B = CH3COOH
(Ground Truth Answer: C)
**TextGrad Response**: ...let's evaluate the options: ... Option B: A = H3O+, B = HCl - H3O+ (hydronium ion) is a suitable proton source for cyanohydrin formation. HCl is a strong acid and commonly used for hydrolyzing nitriles to carboxylic acids. Option C: A = NaHSO3, B = HCl **NaHSO3 is not typically used in cyanohydrin formation.** HCl is suitable for hydrolyzing nitriles. ... Based on the analysis, the most suitable reagents for the reactions are: A = H3O+ for the cyanohydrin formation. B = HCl for the hydrolysis of the nitrile to a carboxylic acid. Therefore, the correct answer is: Answer: B.
**REVOLVE Response**: ...The first reaction is the addition of cyanide ion (CN-) to butan-2-one to form 2-hydroxy-2-methylbutanenitrile. This is an example of a cyanohydrin reaction, which typically involves the addition of cyanide ion to an aldehyde or ketone in the presence of a catalyst such as sodium bisulfite (NaHSO3).The second reaction is the hydrolysis of 2-(4-benzylphenyl)-2-hydroxybutanenitrile to form 2-(4-benzylphenyl)-2-hydroxybutanoic acid. This is an example of a hydrolysis reaction, which typically involves the addition of water (H2O) in the presence of a strong acid such as hydrochloric acid (HCl). **Based on this analysis, the suitable reagents for the reactions are:A = NaHSO3 (sodium bisulfite) for the first reaction. B = HCl (hydrochloric acid) for the second reaction.** So, the correct answer is: Answer: C.

---

**Case Study:** In this example, TextGrad incorrectly selects H3O+ for cyanohydrin formation, misunderstanding the role of NaHSO3, which is a typical catalyst in such reactions. Its reliance on first-order feedback leads to an erroneous conclusion. In contrast, REVOLVE correctly identifies NaHSO3 as the catalyst for the first reaction and HCl for the hydrolysis in the second reaction. By leveraging second-order gradients, REVOLVE better captures the complexities of the chemical mechanisms, leading to the correct answer.

# D. Detailed Experiments on Solution Optimization

Table 5: Solution optimization results for various LLM, with themselves as the optimization engine. The values in parentheses represent the relative improvement of the method compared to TextGrad.

| Dataset | Models | Stage | Accuracy % (Improv. over TextGrad) | | | |
|---|---|---|---|---|---|---|
| | | | CoT | TextGrad | M-TextGrad | REVOLVE |
| Google-proof QA | GPT-4o | Before Training | 50.4 (0.0%) | 50.4 (-) | 50.4 (0.0%) | 50.4 (0.0%) |
| | | 1st Iteration | - | 50.4 (-) | **51.1 (1.3%↑)** | 50.9 ± 0.4% (0.99%↑) |
| | | 2nd Iteration | - | 50.5 (-) | 50.7 (0.4%↑) | **51.3 ± 0.5% (1.58%↑)** |
| | | 3rd Iteration | - | 50.5 (-) | 51.9 (2.7%↑) | **52.9 ± 0.6% (4.75%↑)** |
| | | Final Results | 50.4 (2.1%↓) | 51.5 (-) | 52.4 (1.7%↑) | **53.0 ± 0.7% (2.91%↑)** |
| | GPT-4-0125-preview | Before Training | 38.8 (0.0%) | 38.8 (-) | 38.8 (0.0%) | 38.8 (0.0%) |
| | | 1st Iteration | - | 38.5 (-) | 39.3 (2.0%↑) | **39.5 ± 0.3% (2.60%↑)** |
| | | 2nd Iteration | - | 38.3 (-) | 40.1 (4.7%↑) | **40.3 ± 0.5% (5.22%↑)** |
| | | 3rd Iteration | - | 38.2 (-) | 40.4 (5.7%↑) | **41.0 ± 0.4% (7.33%↑)** |
| | | Final Results | 38.8 (1.8%↑) | 38.1 (-) | 41.5 (8.9%↑) | **42.2 ± 0.6% (10.76%↑)** |
| | Llama 3.1 8B Instruct | Before Training | 21.7 (0.0%) | 21.7 (-) | 21.7 (0.0%) | 21.7 (0.0%) |
| | | 1st Iteration | - | 25.8 (-) | 26.5 (2.7%↑) | **26.8 ± 0.2% (3.88%↑)** |
| | | 2nd Iteration | - | 26.8 (-) | 29.3 (9.3%↑) | **29.8 ± 0.5% (11.19%↑)** |
| | | 3rd Iteration | - | 24.8 (-) | 25.7 (3.6%↑) | **27.8 ± 0.5% (12.10%↑)** |
| | | Final Results | 21.7 (8.4%↓) | 23.7 (-) | 25.1 (5.9%↑) | **28.3 ± 0.4% (19.41%↑)** |
| MMLU-Machine Learning | GPT-4o | Before Training | 85.5 (0.0%) | 85.5 (-) | 85.5 (0.0%) | 85.5 (0.0%) |
| | | 1st Iteration | - | 85.5 (-) | 85.5 (0.0%) | **85.8 ± 0.5% (0.35%↑)** |
| | | 2nd Iteration | - | 85.6 (-) | 85.4 (0.2%↓) | **86.1 ± 1.0% (0.58%↑)** |
| | | 3rd Iteration | - | 85.6 (-) | 85.3 (0.3%↓) | **86.4 ± 1.1% (0.93%↑)** |
| | | Final Results | 85.5 (0.3%↓) | 85.8 (-) | 85.0 (0.9%↓) | **86.7 ± 0.8% (1.05%↑)** |
| | GPT-4-0125-preview | Before Training | 76.3 (0.0%) | 76.3 (-) | 76.3 (0.0%) | 76.3 (0.0%) |
| | | 1st Iteration | - | 76.4 (-) | **77.2 (1.0%↑)** | 77.1 ± 0.4% (0.92%↑) |
| | | 2nd Iteration | - | 76.6 (-) | 77.8 (1.5%↑) | **77.9 ± 0.6% (1.70%↑)** |
| | | 3rd Iteration | - | 77.0 (-) | 78.1 (1.4%↑) | **79.2 ± 0.7% (2.86%↑)** |
| | | Final Results | 76.3 (3.3%↓) | 78.9 (-) | 79.2 (0.3%↑) | **81.0 ± 0.8% (2.66%↑)** |
| | Llama 3.1 8B Instruct | Before Training | 51.8 (0.0%) | 51.8 (-) | 51.8 (0.0%) | 51.8 (0.0%) |
| | | 1st Iteration | - | 43.8 (-) | 46.9 (7.1%↑) | **48.2 ± 0.3% (10.05%↑)** |
| | | 2nd Iteration | - | 43.8 (-) | 45.2 (3.2%↑) | **47.3 ± 0.5% (7.99%↑)** |
| | | 3rd Iteration | - | 43.8 (-) | 44.4 (1.4%↑) | **46.4 ± 0.6% (5.94%↑)** |
| | | Final Results | 51.8 (9.5%↑) | 47.3 (-) | 47.4 (0.2%↑) | **57.1 ± 0.6% (20.72%↑)** |
| MMLU-College Physics | GPT-4o | Before Training | 91.0 (0.0%) | 91.0 (-) | 91.0 (0.0%) | 91.0 (0.0%) |
| | | 1st Iteration | - | 91.6 (-) | 91.6 (0.0%) | **91.8 ± 0.5% (0.22%↑)** |
| | | 2nd Iteration | - | 92.1 (-) | 92.3 (0.2%↑) | **92.5 ± 0.7% (0.43%↑)** |
| | | 3rd Iteration | - | 92.8 (-) | 91.2 (1.7%↓) | **93.2 ± 0.8% (0.43%↑)** |
| | | Final Results | 91.0 (2.7%↓) | 93.5 (-) | 91.3 (2.3%↓) | **94.1 ± 0.9% (0.64%↑)** |
| | GPT-4-0125-preview | Before Training | 81.6 (0.0%) | 81.6 (-) | 81.6 (0.0%) | 81.6 (0.0%) |
| | | 1st Iteration | - | 82.4 (-) | 81.9 (0.6%↓) | **82.5 ± 0.3% (0.12%↑)** |
| | | 2nd Iteration | - | 83.1 (-) | 82.4 (0.8%↓) | **83.4 ± 0.4% (0.36%↑)** |
| | | 3rd Iteration | - | 84.1 (-) | 82.1 (2.4%↓) | **84.5 ± 0.6% (0.48%↑)** |
| | | Final Results | 81.6 (4.7%↓) | 85.6 (-) | 82.3 (3.8%↓) | **85.9 ± 0.7% (0.35%↑)** |
| | Llama 3.1 8B Instruct | Before Training | 54.7 (0.0%) | 54.7 (-) | 54.7 (0.0%) | 54.7 (0.0%) |
| | | 1st Iteration | - | 51.1 (-) | 55.9 (9.4%↑) | **58.3 ± 0.2% (14.09%↑)** |
| | | 2nd Iteration | - | 51.1 (-) | 61.0 (19.4%↑) | **62.0 ± 0.4% (21.33%↑)** |
| | | 3rd Iteration | - | 55.7 (-) | 60.3 (8.3%↑) | **65.7 ± 0.5% (17.95%↑)** |
| | | Final Results | 54.7 (9.3%↓) | 60.3 (-) | 61.6 (2.2%↑) | **66.4 ± 0.5% (10.12%↑)** |

# E. Code Optimization

In the code optimization task, we primarily rely on the settings from previous work, particularly TextGrad(Yuksekgonul et al., 2024), to ensure a fair comparison across experiments. Specifically, we adopt the version of Reflexion(Shinn et al., 2024) as used in TextGrad, which includes minor modifications for compatibility within the TextGrad framework.

For the baselines, we employ two key approaches:

- Reflexion (Shinn et al., 2024): In this setup, the language model is guided by a one-shot prompt instructing it to provide feedback on the code it generates. The process begins by generating an initial solution based on a provided code prompt.

The solution is first tested locally, and if it passes, it is then submitted to the LeetCode platform for a more rigorous evaluation using harder test cases. If the local tests fail, Reflexion is used to request feedback from the model to refine the code. This feedback-guided optimization is repeated for up to 5 iterations, during which the model continuously improves the code until it successfully passes local tests and is ready for submission.

- TextGrad (Yuksekgonul et al., 2024): This baseline runs 5 independent trials, each with five seeds with [15, 17, 21, 55, 91], and averages the results to ensure consistency. During each optimization iteration, TextGrad performs three key operations: first, it makes a call to GPT-4o to evaluate the test time loss; next, it collects gradients based on the loss; finally, the code snippet is updated according to the gradients. This process repeats, optimizing the code over several iterations to minimize the test time loss and improve performance on the test cases.

By following the setup from TextGrad, we ensure that both Reflexion and TextGrad are evaluated under the same conditions, facilitating a fair and consistent comparison with these two baselines and REVOLVE.

---

**Example Query for LeetCode Hard**

def minimumTime(grid: List[List[int]]) -> int:
"""

You are given a 'm x n' matrix 'grid' consisting of non-negative integers where 'grid[row][col]' represents the minimum time required to be able to visit the cell '(row, col)', which means you can visit the cell '(row, col)' only when the time you visit it is greater than or equal to 'grid[row][col]'.
You are standing in the top-left cell of the matrix in the '0th' second, and you must move to any adjacent cell in the four directions: up, down, left, and right. Each move you make takes 1 second. Return the minimum time required in which you can visit the bottom-right cell of the matrix. If you cannot visit the bottom-right cell, then return '-1'.
Example 1:
Input: grid = [[0,1,3,2],[5,1,2,5],[4,3,8,6]]
Output: 7
Explanation:
One of the paths that we can take is the following:
- at t = 0, we are on the cell (0,0).
- at t = 1, we move to the cell (0,1). It is possible because grid[0][1] <= 1.
- at t = 2, we move to the cell (1,1). It is possible because grid[1][1] <= 2.
- at t = 3, we move to the cell (1,2). It is possible because grid[1][2] <= 3.
- at t = 4, we move to the cell (1,1). It is possible because grid[1][1] <= 4.
- at t = 5, we move to the cell (1,2). It is possible because grid[1][2] <= 5.
- at t = 6, we move to the cell (1,3). It is possible because grid[1][3] <= 6.
- at t = 7, we move to the cell (2,3). It is possible because grid[2][3] <= 7.
The final time is 7. It can be shown that it is the minimum time possible.
Example 2:
Input: grid = [[0,2,4],[3,2,1],[1,0,4]]
Output: -1
Explanation:
There is no path from the top left to the bottom-right cell. Constraints:
* 'm == grid.length'
* 'n == grid[i].length'
* '2 <= m, n <= 1000'
* '4 <= m * n <= 105'
* '0 <= grid[i][j] <= 105'
* 'grid[0][0] == 0'
"""

---

---

Example Comparison of TextGrad and REVOLVE

**Task Description:**
"""

You have $k$ bags. You are given a **0-indexed** integer array $weights$ where $weights[i]$ is the weight of the $i^{th}$ marble. You are also given the integer $k$.

Divide the marbles into the $k$ bags according to the following rules:

- No bag is empty.
- If the $i^{th}$ marble and $j^{th}$ marble are in a bag, then all marbles with an index between the $i^{th}$ and $j^{th}$ indices should also be in that same bag.
- If a bag consists of all the marbles with an index from $i$ to $j$ inclusively, then the cost of the bag is $weights[i] + weights[j]$.

The score after distributing the marbles is the sum of the costs of all the $k$ bags.

Return the **difference** between the **maximum** and **minimum** scores among marble distributions.

Example 1:

Input: weights = [1,3,5,1], k = 2

Output: 4

Explanation:

The distribution [1],[3,5,1] results in the minimal score of (1+1) + (3+1) = 6.

The distribution [1,3],[5,1], results in the maximal score of (1+3) + (5+1) = 10.

Thus, we return their difference 10 - 6 = 4.

Example 2:

Input: weights = [1, 3], k = 2

Output: 0

Explanation:

The only distribution possible is [1],[3].

Since both the maximal and minimal score are the same, we return 0.
"""

def putMarbles(weights: List[int], k: int) -> int:

**Code Generated by TextGrad**: from typing import List

def putMarbles(weights: List[int], k: int) -> int: n = len(weights) if k == 1: return 0

Calculate the cost of each possible bag costs = [weights[i] + weights[i + 1] for i in range(n - 1)]

Sort the costs to find the minimum and maximum possible scores costs.sort()

The minimum score is the sum of the smallest k-1 costs $min_s core = sum(costs[: k - 1])$

The maximum score is the sum of the largest k-1 costs $max_s core = sum(costs[-(k - 1) :])$

return $max_s core - min_s core$ **Tests that the code passed:** No tests passed.

**Tests that the code failed:**

assert putMarbles([1,3,5,1], 2) == 4 ERROR: This unit test fails. Output was 2, but expected value was: 4 assert putMarbles([1,3], 2) == 0 ERROR: This unit test fails. Output was -1, but expected value was: 0

**Code Generated by REVOLVE** from typing import List

def putMarbles(weights: List[int], k: int) -> int: n = len(weights) if k == 1 or k == n: return 0

Calculate the cost of each possible bag costs = [weights[i] + weights[i + 1] for i in range(n - 1)]

Sort the costs to find the minimum and maximum possible scores costs.sort()

The minimum score is the sum of the smallest k-1 costs $min_s core = sum(costs[: k - 1])$

The maximum score is the sum of the largest k-1 costs $max_s core = sum(costs[-(k - 1) :])$

return $max_s core - min_s core$

**Tests that the code passed:** assert putMarbles([1,3,5,1], 2) == 4 assert putMarbles([1,3], 2) == 0

**Tests that the code failed:**

No tests failed.

# F. Abaltion Study

## F.1. Ablation Study

Given the simplicity of our method, there are no complex components that can be eliminated for a traditional ablation. Instead, we conduct an ablation study by testing different prompt designs to evaluate their impact on performance. Specifically, using Llama-3.1-8B-Instruct as the LLM backend, we compare our REVOLVE prompt, a variant of this prompt, and the prompt used in TextGrad to highlight the effectiveness of our approach.

- **REVOLVE Prompt:** This is the prompt carefully designed for REVOLVE, which takes into account both immediate feedback and the evolution of responses across iterations.
- **Variant Prompt:** It differs from our method by directly instructing the LLM to generate more diverse responses, effectively pushing it towards greater variation with each iteration.
- **TextGrad Prompt:** Serving as the baseline, TextGrad's prompt focuses primarily on immediate feedback, making adjustments based solely on the latest response.

The detailed prompts for the variant are as follows:

---

**OPTIMIZER SYSTEM PROMPT**

"You are part of an optimization system that improves text (i.e., variable). "
"You will be asked to creatively and critically improve prompts, solutions to problems, code, or any other text-based variable. "
"You will receive some feedback, and use the feedback to improve the variable. "
"Pay attention to the role description of the variable, and the context in which it is used. "
"Importantly, focus on creating responses that are varied and diverse in nature. "
"You MUST give your response by sending the improved variable between {new_variable_start_tag} {{improved variable}} new_variable_end_tag tags. "
"The text you send between the tags will directly replace the variable."
f"{GLOSSARY_TEXT}"

---

**Textual Gradient Descent Prompt Prefix**

"Here is the role of the variable you will improve: <ROLE>{variable_desc}</ROLE>."
"The variable is the text within the following span: <VARIABLE> {variable_short} </VARIABLE>"
"Here is the context and feedback we got for the variable:"
"<CONTEXT>{variable_grad}</CONTEXT>"
"Improve the variable ({variable_desc}) using the feedback provided in <FEEDBACK> tags. "
"Ensure that your response introduces new and diverse ways of solving the problem or addressing the prompt."

---

The following is how we save gradients to the variable.

---

**GRADIENT TEMPLATE**

"Here is a conversation:<CONVERSATION>{context}</CONVERSATION>"
"This conversation is part of a larger system, where varied and creative outputs are important. "
"The output is used as {response_desc}. Here is the feedback we received for {variable_desc} in the conversation:"
"<FEEDBACK>{feedback}</FEEDBACK>"
"Encourage diversity in your improvements."

---

We conducted experiments on objective counting in the prompt optimization task, with results shown in Table 6.

On the Object Counting task, the Variant prompt surpasses TextGrad by encouraging larger, more diverse shifts in the response space, enabling the model to explore more distinct outputs with each iteration. REVOLVE, on the other hand,

Table 6: Prompt optimization results for reasoning tasks for various LLMs, with gpt-4o as the optimization engine.

| Dataset | Method | Accuracy % |
|---|---|---|
| Object Counting | TextGrad | 77.0 |
| | Variant | 80.0 |
| | REVOLVE | **83.0** |

Table 7: Comparison of computational resources (GPU memory and runtime) for REVOLVE and baseline methods across tasks.

| Task | Dataset | Method | Time per Iteration (s) | Total Time to Converge (s) | GPU Usage (GB) |
|---|---|---|---|---|---|
| Prompt Optimization | Objective Counting | TextGrad | 92.144 | 276.450 | 3.23 |
| | | M-TextGrad | 110.721 | 110.732 | 3.24 |
| | | REVOLVE | 137.815 | 137.821 | 3.23 |
| | GSM8K | TextGrad | 135.184 | 1351.85 | 3.23 |
| | | M-TextGrad | 152.423 | 1219.393 | 3.23 |
| | | REVOLVE | 176.538 | 1235.774 | 3.23 |
| Solution Optimization | Google-proof QA | TextGrad | 153.522 | 614.216 | 3.24 |
| | | M-TextGrad | 178.879 | 1091.162 | 3.23 |
| | | REVOLVE | 197.235 | 453.461 | 3.24 |
| | MMLU-Machine Learning | TextGrad | 172.429 | 896.631 | 3.24 |
| | | M-TextGrad | 207.819 | 685.803 | 3.24 |
| | | REVOLVE | 223.807 | 626.659 | 3.24 |
| | MMLU-College Physics | TextGrad | 188.116 | 1636.612 | 3.24 |
| | | M-TextGrad | 225.631 | 1308.662 | 3.24 |
| | | REVOLVE | 245.167 | 1054.229 | 3.24 |
| Code Optimization | Objective Counting | TextGrad | 1078.783 | 18986.655 | 6.46 |
| | | M-TextGrad | 1241.917 | 18472.411 | 6.46 |
| | | REVOLVE | 1352.174 | 15820.489 | 6.46 |

achieves even better results by promoting stable, iterative refinement rather than abrupt changes. While the Variant's strategy can lead to sudden, exaggerated shifts, REVOLVE ensures smoother, controlled optimization, gradually fine-tuning responses for greater accuracy.

## G. Comparison of Computational Resources

To analyze the computational efficiency of REVOLVE, we compare its GPU memory usage and runtime against baseline methods across three task categories. We use Llama 3.1 8B Instruct as the base LLM, running on a setup with 4 NVIDIA 3090 GPUs. The results are shown in Table 7.

We observe that while REVOLVE involves slightly higher per-iteration runtime due to its second-order optimization-inspired design, it converges in fewer iterations, resulting in significant overall savings. The detailed findings are as follows:

- On Object Counting dataset, REVOLVE reduces total runtime by 50% compared to TextGrad by converging in fewer iterations despite slightly higher per-iteration costs.
- For solution optimization task, REVOLVE achieves 26.14% lower total runtime than TextGrad, while M-TextGrad incurs 77.65% higher runtime due to instability.
- For code optimization task, REVOLVE reduces total runtime by 16.67% compared to baselines.
- For GPU memory usage, REVOLVE demonstrates similar requirements to baseline methods, indicating no significant increase in computational resources.

