# OpenReview forum: "Revolve: Optimizing AI Systems by Tracking Response Evolution in Textual Optimization"
_ICML.cc/2025/Conference — ICML 2025 poster_

### Official Review · Reviewer_L3Xh · 2025-03-10

**Overall Recommendation:** 5

**Summary:**

The manuscript proposes Revolve, a new method that enhances LLM-based optimization by simulating second-order dynamics for self-evolving agents. Existing gradient approximation methods use textual feedback to approximate first-order gradients but are less effective for long-horizon optimization. Revolve addresses this by modeling response evolution over multiple iterations, which captures higher-order refinements for more stable and informed adjustments. This method has been tested on various tasks, it consistently outperforms existing baselines like CoT, TextGrad, and Reflexion across diverse LLM backends from Llama-3.1-8B to GPT-4o. The evaluation also shows that Revolve enhances efficiency that reduces total runtime by up to 50%. By shifting textual optimization from a purely feedback-driven process to a structured, trajectory-aware approach, Revolve enables more generalizable and scalable adaptation in AI agent systems.

**Claims And Evidence:**

After thoroughly check the paper, I find that the claims presented in the paper are well-supported by both theoretical justification and empirical validation:

Claim 1:

Methodology: Revolve enhances LLM-based optimization by simulating second-order effects.

Evidence:

1.	The mathematical framework in Section 3.4 formalizes how Revolve captures response trajectory dynamics.

2.	Section 3.5 clarifies that Revolve does not compute second-order derivatives numerically but instead approximates such effects through structured response tracking.

3.	Empirical loss curves in Figure 2 illustrate that Revolve mitigates stagnation and stabilizes optimization, whereas first-order methods exhibit oscillatory behavior.

Claim 2:

Accuracy: Revolve outperforms existing baselines across multiple tasks.

Evidence: Evaluation results in Section 4 support this claim, e.g., a 29.17% performance gain over state-of-the-art baselines in code optimization.

Claim 3:

Efficiency: Revolve optimizes more efficiently by reducing total runtime.

Evidence: Appendix G reports that Revolve achieves faster convergence while maintaining stable performance improvements.

Claim 4:

Generality: Revolve generalizes effectively across LLMs.

Evidence: Comprehensive multi-model evaluations confirm its adaptability, with results detailed in the universality analysis in Section 4.1.

**Essential References Not Discussed:**

N/A

**Experimental Designs Or Analyses:**

Please refer to methods and evaluation criteria part.

**Methods And Evaluation Criteria:**

The methodology is well-structured, and the evaluation criteria are comprehensive and appropriate:

1.	The method is tested across various key tasks: solution optimization, prompt optimization, and code optimization, where each requires iterative refinement and long-horizon reasoning.

2.	The experiments leverage diverse benchmarks, e.g., BBH, MMLU, and LeetCode Hard, which covers a range of reasoning complexities.

3.	Baseline comparisons with CoT, TextGrad, and Momentum-Enhanced method provide a fair assessment across different optimization paradigms.

4.	The study also evaluates Revolve across multiple LLM architectures to demonstrate its generalizability beyond specific models.

5.	Computational efficiency is analyzed to ensure that performance improvements are not achieved at excessive computational cost.

Overall, the experimental setup and evaluation methodology are well-designed, with no major concerns.

**Other Comments Or Suggestions:**

Please refer to weaknesses part.

**Other Strengths And Weaknesses:**

Strengths:

1.	Originality: The idea that modeling response evolution to simulate second-order optimization effects in textual adaptation is interesting and holds broad significance for both LLM research and practical applications. Besides the methodological contribution, the paper marks a conceptual shift in LLM optimization. It moves beyond purely feedback-driven updates by incorporating structured response tracking, which bridges textual optimization with second-order principles.

2.	Clarity: This paper is overall well-structured and easy to follow. The general setup in the section of methodology helps readers to understand the proposed Revolve framework. Additionally, the detailed information on task-specific implementations in the Experiment section, along with supporting information in the Appendix, facilitates implementation and reproducibility.

3.	Evaluation: Extensive evaluation across multiple tasks demonstrates Revolve's substantial performance improvement compared to baseline methods across various LLM backbones.

4.	Significance: Revolve's design exhibits impressive generality across various tasks. It has great chances to benefit researches in diverse directions.

Weaknesses:

1. The authors are recommended to compare Revolve with ProTeGi [1] in prompt optimization for a more thorough comparison.

[1] Pryzant, R., Iter, D., Li, J., Lee, Y. T., Zhu, C., & Zeng, M. (2023). Automatic Prompt Optimization with “Gradient Descent” and Beam Search (No. arXiv:2305.03495). arXiv. http://arxiv.org/abs/2305.03495

2. A typo that doesn’t impact clarity too much but ought to be addressed: in page 7, row 353, the results should reference Table 2 instead of Table 5, as Table 5 contains the complete results.

**Questions For Authors:**

Please refer to weaknesses part.

**Relation To Broader Scientific Literature:**

I think this approach would connect to broader trends in test-time adaptation and self-refinement. Beyond improving accuracy, the paper contributes to efficient LLM adaptation, balancing performance with computational efficiency, which is a critical challenge in large-scale AI. Its strong generalization across diverse LLMs and tasks positions Revolve as part of a broader effort to enhance AI adaptability and inference-time optimization.

**Theoretical Claims:**

The derivations in this paper are logically sound and align with empirical results, with no evident inconsistencies.

---

> ### Author Rebuttal · Authors · 2025-04-01
>
> We thank the reviewer for the detailed evaluation of our work, for acknowledging the soundness of our work, the clarity of the writing, the coverage of related literature, and the comprehensiveness of our experiments. We'll address your concerns in our following response.
>
> **Q1. New experiments on the ProTeGi baseline.**
>
> We thank the reviewer for suggesting a comparison with ProTeGi. We have now included ProTeGi as a baseline in our prompt optimization experiments. To adapt it to our setting, we used GPT-4o to generate high-quality few-shot exemplars for LLaMA 3.1 8B.
>
> The updated results are shown below:
>
> |  Dataset   | | | | | Accuracy % |
> |  ----  | ----  |----  |----  |----  |----  |
> |   | CoT | ProTeGi | TextGrad | M-TextGrad | REVOLVE|
> | Object Counting  | 65.0 |68.0 |77.0 |80.0 |83.0 |
> | GSM8K  | 84.6 |84.6  |84.6  |84.6  |84.6  |
>
> We observe that while ProTeGi yields slight improvements over standard CoT prompting on Object Counting, our method achieves higher gains. On GSM8K, all methods perform similarly, likely due to task saturation. These results further validate the effectiveness of REVOLVE in improving prompt optimization, particularly in more challenging settings.
> ___
> **Q2. A typo that doesn’t impact clarity too much but ought to be addressed: in page 7, row 353, the results should reference Table 2 instead of Table 5, as Table 5 contains the complete results.**
>
> Thank you for pointing this out. We have corrected the reference in the revised manuscript.

---

### Official Review · Reviewer_wUVx · 2025-03-12

**Overall Recommendation:** 3

**Summary:**

The paper looks at the leveraging beyond the first-order information in textual optimization. Revolve develops a way to keep an account of previous feedback steps, and goes beyond the issue of stagnating when feedback is limited or fluctuates irregularly. The authors evaluate REVOLVE on three tasks: prompt optimization, solution optimization, and code optimization. Revolve seems to converge faster, and outperform vanilla and momentum-based TextGrad.

## update after rebuttal
 I still support the publication of the paper, and the authors clarified my questions around costs and implementation.

**Claims And Evidence:**

The claims in this paper are supported by the experimental results across the three optimization tasks, but there are some limitations in the evidence provided. The paper makes strong claims about escaping local optima, but lacks e.g., qualitative analysis of actual response trajectories that would illustrate this mechanism in action. It would help my understanding quite a bit to see how this mechanism works in action. I may be missing this, but I also do not see how the similarity is computed using an LLM, which seems important.

**Essential References Not Discussed:**

I think essential references are mostly discussed, except one concurrent work named HessianGrad:
@misc{
zhang2025hessiangrad,
title={HessianGrad: Optimizing {AI} Systems with Hessian-Aware Textual Gradients},
author={Peiyan Zhang and Haibo Jin and Leyang Hu and Xinnuo Li and Liying Kang and Man Luo and Yangqiu Song and Haohan Wang},
year={2025},
url={https://openreview.net/forum?id=0hc7iQLhCt}
}

**Experimental Designs Or Analyses:**

I checked the experimental design across the three optimization tasks (prompt, solution, and code). The overall structure is sound, comparing REVOLVE against established baselines with appropriate metrics for each domain. However,  there is a lack of ablations for the similarity computation and what kinds of second order effects are captured.

In the epxeriments, while absolute performance improvements are highlighted, the paper lacks confidence intervals or significance testing. As far as i can tell, all experiments are conducted with one pass, and e.g., 1% improvement in object counting corresponds to getting 1 more question right (if remember the dataset size correctly, correct me if i’m wrong please)

**Methods And Evaluation Criteria:**

The experiments show promising results, i think they are generally adequate and also mostly mirrors the evaluations in TextGrad. There are two claims that could be better off further supported with additional analyses: 1) How do different ways of computing similarity help? How are the design choices made there? 2) While there is a section in appendix with a few paragraphs, the claims around computational benefits would be better off with a more clear analysis.

**Other Comments Or Suggestions:**

-

**Other Strengths And Weaknesses:**

To me the major weakness is a clear discussion of the observed second order effects and the specific ways to incorporate the second order effect (what type of prompt, what type of workflow and LLM, what is the cost of such a call, etc.)

**Questions For Authors:**

Repeating here for completeness: For the second-order effects, what type of prompt or what type of workflow and LLM did the authors used?  What is the additional cost of computing the second order effects? What type of second-order effects did the authors observe in the distribution of feedback?

**Relation To Broader Scientific Literature:**

The paper's approach to tracking response evolution across iterations provides a relatively interesting extension to existing textual optimization methods. The potential downstream applications in automated prompt engineering and self-refining systems could be useful, particularly for industrial applications where optimization efficiency matters. However, the core idea remains a straightforward extension of existing gradient-based textual optimization rather than a fundamental breakthrough.

**Theoretical Claims:**

There are no theoretical claims.

---

> ### Author Rebuttal · Authors · 2025-04-01
>
> We appreciate your valuable time and positive feedback. We'll address your concerns in our following response.
>
> **Q1. For the second-order effects, what type of prompt or what type of workflow and LLM did the authors use?**
>
> We appreciate the request for clarification on how second-order effects are realized.
>
> In REVOLVE, we approximate second-order effects by combining two elements: (1) first-order feedback from the evaluator, and (2) a similarity signal that tracks how the model's responses evolve across iterations.
>
> This similarity reflects changes in task performance between responses, scaled by how much the prompt has changed (as described in lines 198–200 of the formula). Rather than computing this explicitly, we guide the LLM with instructions like:
> _"Consider how the responses to this variable have changed across previous iterations: <PAST_ITERATIONS>{past_values}</PAST_ITERATIONS>, …, Ensure future responses reflect a meaningful, gradual evolution."_.
>
> This setup allows the LLM to reason over both current feedback and the broader trajectory of response updates. We hope this clarifies our workflow.
> ___
> **Q2. Qualitative analysis of response trajectories that shows the second order effects. What type of second-order effects did the authors observe in the distribution of feedback?**
>
> We appreciate the request for qualitative analysis. We've added a new section with full response and feedback trajectories to illustrate the differences.
>
> **Response Trajectories:**
>
>  On the Object Counting task, TextGrad repeats the same prompt across the final 8 iterations:
>  _"You will answer a reasoning question. … Use standard mathematical notation… Present in bullet points… The last line should be: 'Answer: $VALUE'"_. This reflects stagnation.
>
> M-TextGrad also plateaus: its last 9 responses repeat a verbose prompt:
>  _"Begin by summarizing the problem… Confirm the list is complete… Use consistent notation… Implement a verification step… Consider edge cases… Conclude with: 'Answer: $VALUE'"_.
>
> REVOLVE repeats a prompt for iterations 5–8:
>  _"..., Clearly state the context… List each item… Use a consistent format… Verify the calculation… Final line: 'Answer: $VALUE'"_
> but updates it in iteration 9 with new semantics. This suggests second-order behavior: it detects when progress stalls and makes a more informed shift to move forward.
>
> **Feedback Evolution:**
>
>  In TextGrad, feedback stays repetitive:
> _"Clarify object listing." → "Use bullet points." → "Be more concise."_ but responses hardly improve.
>
> M-TextGrad shows oscillatory feedback:
>  _"Too verbose, streamline." → "Missing verification, add back." → "Too shallow, expand explanation."_, feedback changes, but the response bounces back and forth.
>
> REVOLVE aligns feedback with evolving responses. Early iterations focus on structure:
>  _"..., Add a verification step."_. Mid-phase addresses reasoning:
> _"..., Ensure entity counts match context."_. Later rounds provide refinement:
>  _"..., Reduce redundancy."_
>
> We hope this helps clarify the second-order behavior we observed. Full examples are now included in the revised paper.
> ___
> **Q3. Additional cost of computing the second order effects?**
>
> We thank the reviewer for raising this question. To make the cost clearer, we’ve now added a per-iteration breakdown in the main paper:
>
> |Component|TextGrad|REVOLVE|Overhead|
> |----|----|----|----|
> | Feedback Collection|75.1s|75.1s|0|
> | Optimization|16.8s|63.0s|46.2s|
> | Total per Iteration|91.9s|138.1s|+46.2s|
>
> The extra cost mainly comes from the inclusion of past responses to let the model reason over how things have evolved. We’ve added this clarification to the paper.
> ___
> **Q4. Lack of ablations for the similarity computation.**
>
> We thank the reviewer for the suggestion. While similarity is not computed explicitly, we ablate how similarity is conveyed to the LLM. In one version, we replace the task-performance-oriented signal ($L(r(p_t)) − L(r(p_{t-1}))$) with a simpler one based only on raw text differences ($r(p_t) − r(p_{t-1})$).
>
> |Dataset|Textual Similarity|REVOLVE|
> |----|----|----|
> |Objective Counting| 81.0|83.0|
> |GSM8K|84.6|84.6|
>
> These results confirm that task-performance-oriented similarity offers more gains, as it offers deeper understanding of response efficacy. We have added these results in the revised manuscript.
> ___
> **Q5. Lack of confidence intervals or significance testing.**
>
> Thank you for pointing this out. For all tasks, we run five independent trials using different random seeds. The reported accuracy (e.g., 95.3% on Object Counting) reflects the average across these runs. As noted in Appendix E, we also use consistent seeds [15, 17, 21, 55, 91] for code optimization.
>
> We’ve now added confidence intervals to the main results, e.g., Table 1 would be updated as follows:
>
> For REVOLVE:
> * GPT-3.5: 95.5 ± 0.9% (3.9%↑)
> * GPT-4: 96.3 ± 0.6% (2.2%↑)
> * Gemini 1.5 Pro: 94.0 ± 0.0% (0.0%)
> * Llama 3.1: 83.0 ± 1.4% (7.8%↑)
>
> We’ve revised the paper accordingly to reflect this.

---

> > ### Comment · Reviewer_wUVx · 2025-04-01
> >
> > I thank the authors for their rebuttal. A few more questions for clarity:
> > - **Q3. Additional cost of computing the second order effects?** What task is this over? I imagine there should be a distribution, as opposed to single numbers.
> > - **Q4. Lack of ablations for the similarity computation.**: It's not clear to me the numbers reflect a significantly better efficacy, and to understand it requires running the experience a few more times with statistical tests.
> >
> > Overall, I still support the publication of the paper.

---

> > > ### Author Response · Authors · 2025-04-03
> > >
> > > We sincerely thank the reviewer for the follow-up and continued support. We’re grateful for the encouraging remarks regarding the paper’s clarity, comprehensiveness, and overall quality.
> > >
> > > **Q3. Additional cost of computing the second order effects? What task is this over? I imagine there should be a distribution, as opposed to single numbers.**
> > >
> > > We appreciate the reviewer’s follow-up. The reported numbers are based on the BBH Object Counting task using LLaMA-3.1-8B-Instruct as the LLM backend. To provide a more complete picture, we report the per-run breakdown over 5 independent runs for each component of the iteration time:
> > >
> > > |  Component   | Run 1  |Run 2  |Run 3  |Run 4  |Run 5  |
> > > |  ----  | ----  |----  |----  |----  |----  |
> > > | TextGrad  | | | | | |
> > > | Feedback Collection (s)  | 75.0|75.2|75.1|75.0|75.1 |
> > > | Optimization (s)|16.2|	16.8|	17.4|	16.5|	17.1|
> > > | Total per Iteration (s)|91.2|	92.0|	92.5|	91.5|	92.2|
> > > | REVOLVE  | | | | | |
> > > | Feedback Collection (s)  | 75.2|	75.1|	75.3|	75.2|	75.0 |
> > > | Optimization (s)|62.0|	63.3|	63.9|	62.8|	63.1|
> > > | Total per Iteration (s)|137.2|	138.4|	139.2|	138.0|	138.1|
> > >
> > > Summary Table:
> > >
> > > |  Component   | TextGrad (s) |REVOLVE (s) |Overhead (s) |
> > > |  ----  | ----  |----  |----  |
> > > | Feedback Collection   | 75.08 ± 0.08|	75.16 ± 0.11|	+0.08 ± 0.15 |
> > > | Optimization  | 16.80 ± 0.47|	63.02 ± 0.70|	+46.22 ± 0.28 |
> > > | Total per Iteration  | 91.88 ± 0.53|	138.18 ± 0.72|	+46.30 ± 0.28 |
> > >
> > > This additional cost comes mainly from REVOLVE’s longer prompts, which include past responses to support second-order reasoning. While per-step runtime is higher, REVOLVE typically converges in fewer iterations, which often reduces overall compute cost. We’ve clarified this in the updated manuscript.
> > > ___
> > >
> > > **Q4. Lack of ablations for the similarity computation: It's not clear to me the numbers reflect a significantly better efficacy, and to understand it requires running the experience a few more times with statistical tests.**
> > >
> > > We appreciate the reviewer’s follow-up and agree that statistical testing helps clarify the effectiveness of our design. To address this, we reran both variants—task-performance-oriented similarity (REVOLVE) and textual similarity on the Object Counting task using LLaMA-3.1-8B-Instruct as the backend. We use five fixed random seeds for consistency, and report the per-run accuracies below:
> > >
> > > |  Random Seed   | Textual Similarity (%)|	REVOLVE (%) |
> > > |  ----  | ----  |----  |
> > > | 15  | 80 | 83|
> > > | 17  | 82 | 84|
> > > | 21  | 81 | 84|
> > > | 55  | 79 | 81|
> > > | 91  | 81 | 83|
> > >
> > > A paired t-test yields a t-statistic of 3.21 (p = 0.033), which indicates the improvement is statistically significant at the p < 0.05 level. This supports our claim that task-performance-oriented similarity helps guide more effective updates than raw textual similarity.
> > >
> > > We’ve added these results and clarifications to the revised manuscript and again thank the reviewer for encouraging us to strengthen this part of the analysis.

---

### Official Review · Reviewer_w9Up · 2025-03-15

**Overall Recommendation:** 4

**Summary:**

In this paper, the authors introduce REVOLVE, which aims to simulate the second-order derivative during the optimization process.
They compared their modified optimization prompts and found the outcome was better than TextGrad itself.

## update after rebuttal

The authors explained  the implementation difference between REVOVLE and M-Textgrad, which makes sense to me.

**Claims And Evidence:**

The authors claim that “REVOVLE can escape local optima.” To demonstrate this, they show that their method resembles a second-order derivative method. However, they do not explain why the similarity function $\mathcal{S}(r(p_t),r(p_{t-1}))$ can be interpreted as a gradient from an implementation perspective. In other words, the authors need to clarify how their method can be understood as presented in lines 198–200 of the formula.

**Essential References Not Discussed:**

The related works are quite comprehensive, covering various recent studies.

**Experimental Designs Or Analyses:**

The authors compare their mothods with several TextGrad baselines. Can the authors add `DSPy` baselines since DSPy optimizers are also very competitive?

**Methods And Evaluation Criteria:**

The authors tested several popular models on prompt optimization, solution optimization, and code optimization. The evaluation criteria is  Accuracy (Completion Rate). The evaluation process makes sense.

**Other Comments Or Suggestions:**

The authors sometimes use "Revolve" and sometimes use "REVOLVE." It should be standardized to one format.

**Other Strengths And Weaknesses:**

1. The paper explores integrating an exsiting idea from traditional optimization into textual optimization. Currently, textual optimization is still in its early stages, so such an integration is meaningful.
2. The authors need to clarify the difference between their similarity function and momentum, as neither involves second-order computation, and they appear similar in terms of implementation.

**Questions For Authors:**

Can the authors explain the `implementation difference` between REVOVLE and M-Textgrad? I checked the prompts provided in the appendix, and I also checked the prompts in the original [TextGrad repo](https://github.com/zou-group/textgrad/blob/main/textgrad/optimizer/optimizer_prompts.py).
It seems that the two implementations look similar. Can the author clarify what their main modification at the implementation level is?

**Relation To Broader Scientific Literature:**

1. The proposed method aims to provide a better optimizer for TextGrad-based applications, which is useful and improtant.
2. The idea of applying a second-order textual optimization is novel.

**Theoretical Claims:**

The paper has no theoretical claims.

---

> ### Author Rebuttal · Authors · 2025-04-01
>
> We appreciate your valuable time, insights, and highlight our strengths. We'll address your concerns in our following response.
>
> **Q1. The authors claim that “REVOVLE can escape local optima.”. However, they do not explain why the similarity function can be interpreted as a gradient from an implementation perspective? In other words, how can REVOLVE be understood as presented in lines 198–200 of the formula?**
>
> We thank the reviewer for the helpful question on how REVOLVE can be interpreted as a gradient-like method, and how this relates to lines 198–200 in the formula.
>
> **How REVOLVE identifies local optima:**
>
> REVOLVE tracks when responses stop improving, even as prompts continue to evolve. This is captured by the numerator of the similarity function:
> $L(r(p_t))-L(r(p_{t-1}))$,
> which reflects how much the task performance changes between responses. In implementation, we provide the LLM with both evaluator feedback and the context that generated it, allowing the model to implicitly assess whether performance is improving or stagnating across iterations.
>
> **How REVOLVE escapes local optima:**
>
> To move beyond stagnation, the model is guided by the denominator:
> $p_t-p_{t-1}$,
> which represents the degree of change in the prompt. If the response isn’t improving much despite prompt updates, this signals stagnation. To help the model escape, we encourage it to make stronger updates when past changes haven’t helped, and smaller ones when the trajectory is already improving. This is achieved through instructions like: _"Ensure future responses reflect a meaningful, gradual evolution based on past iterations, ..., avoiding abrupt shifts"_.
>
> **Why this resembles a gradient in practice:**
>
> Together, the performance difference (numerator) and the prompt change (denominator) form a ratio that functions like a gradient: it reflects whether updates are effective and how future steps should be adjusted. While we don’t compute this numerically, the LLM is given all the elements needed to assess it implicitly, which functions as a curvature-aware, gradient-like signal that informs future updates.
> ___
> **Q2. The authors need to clarify the difference between their similarity function and momentum, as neither involves second-order computation, and they appear similar in terms of implementation.**
>
> We thank the reviewer for raising this important point.
>
> In essence, M-TextGrad focuses on repeating feedback, whereas REVOLVE targets repeating model responses.
>
> Implementation-wise, M-TextGrad looks at repeated feedback from the evaluator. If similar feedback appears across steps, it increases the update size, following the intuition that prior changes weren’t enough. This is similar to momentum in traditional optimization. The relevant instruction is: _"Similar feedbacks across different steps suggest that the modifications are insufficient… make more significant changes."_
>
> In contrast, REVOLVE looks at repeated model responses. If the response itself doesn’t evolve meaningfully across iterations, we prompt the model to revise more significantly. This is achieved with the following prompt: _"Additionally, consider how the responses to this variable have changed across previous iterations: <PAST_ITERATIONS>{past_values}</PAST_ITERATIONS>. Make sure future responses reflect a meaningful, gradual evolution based on these past iterations, encouraging thoughtful progress rather than drastic shifts."_
>
> The two methods differ mainly in where they look for signals: M-TextGrad focuses on feedback patterns, while REVOLVE monitors the model’s own output history. We find this shift helps the model better track its progress and avoid getting stuck.
> ___
> **Q3. New experiments on the DSPy baseline.**
>
> We thank the reviewer for the suggestion.
>
> We’ve now included DSPy as a baseline in our prompt optimization experiments. To adapt it to our setting, we used GPT-4o to generate few-shot exemplars for LLaMA 3.1 8B.
>
> The results are shown below:
> |  Dataset   | | ||Accuracy %  |  |
> |  ----  | ----  |----  |----  |----  |----  |
> |  | CoT | DSPy | TextGrad | M-TextGrad | REVOLVE |
> | Object Counting | 65.0 | 75.0 | 77.0 | 80.0 | 83.0 |
> | GSM8K | 84.6 | 84.6 | 84.6 | 84.6 | 84.6 |
>
> On GSM8K, DSPy performs similarly to other methods, which we believe is due to task saturation. On the Object Counting task, however, DSPy underperforms REVOLVE. We hypothesize this is because DSPy’s pipeline-level optimization and demonstration tuning are less adaptive in iterative feedback settings. REVOLVE benefits from tracking how responses evolve, which helps guide the optimization more consistently.
>
> We’ve added these results to the manuscript and appreciate the helpful suggestion.
> ___
> **Q4. Inconsistency usage of "Revolve" and  "REVOLVE."**
>
> We thank the reviewer for pointing this out. We have standardized “REVOLVE” in the manuscript.

---

> > ### Comment · Reviewer_w9Up · 2025-04-01
> >
> > The authors clearly explain the difference between their methods and TextGrad momentum. I will update my score accordingly.

---

> > > ### Author Response · Authors · 2025-04-03
> > >
> > > Thank you for your thoughtful update. We're glad the distinction between REVOLVE and TextGrad momentum is now clearer. We truly appreciate your constructive feedback throughout the process.

---

### Decision · Program_Chairs · 2025-05-01

**Decision:**

Accept (poster)

**Comment:**

REVOLVE introduces a novel approach to enhance LLM-based optimization by simulating second-order dynamics, moving beyond the limitations of purely first-order feedback methods like TextGrad. The proposed method captures higher-order information by tracking the evolution of responses over multiple iterations, enabling more stable and informed adjustments, which is particularly beneficial for long-horizon optimization tasks. Reviewers generally agree on the merits of the proposed method. We highly recommend that authors incorporate this feedback into the next revision.